# Current Challenges and Future Perspectives of Diagnosis of Hepatitis B Virus

**DOI:** 10.3390/diagnostics13030368

**Published:** 2023-01-19

**Authors:** Manoj Kumar, Sangeeta Pahuja, Prashant Khare, Anoop Kumar

**Affiliations:** 1National Institute of Biologicals, Noida 201309, India; 2Department of Immunohaematology and Blood Transfusion, Lady Hardinge Medical College and Associated Hospitals, New Delhi 110001, India; 3Center for Advanced Biotechnology Research, Xenesis Institute, 5th Floor, Plot 68, Sector 44, Gurugram 122003, India

**Keywords:** HBV, hepatitis, diagnosis, prevention, hepatitis B core-related antigen, hepatitis B surface antigen

## Abstract

It is estimated that approximately 260 million people worldwide are infected with the hepatitis B virus (HBV), which is one of the leading causes of liver disease and liver cancer throughout the world. Compared with developed countries, low-income and middle-income countries have limited access to resources and advanced technologies that require highly specialized staff for HBV diagnosis. In spite of the heavy burden caused by hepatitis B virus, 90% of people are still undiagnosed. The World Health Organization (WHO) goal of eliminating hepatitis B by 2030 seems very difficult to achieve due to the existing diagnostic infrastructure in low-resource regions. The majority of diagnostic laboratories still use hepatitis B surface antigen (HBsAg)-based tests. WHO’s elimination plan is at risk of derailment due to phases like the window period, immune control, and occult HBV infection (OBI) not being detected by standard tests. Here, in this article, we are focusing on various diagnostic platforms for the better diagnosis of HBV. The aim of the elimination of HBV can only be achieved by detecting all phases of HBV infection, which can be executed by a combined approach of using new marker assays along with advanced pretesting and testing methods.

## 1. Introduction

Hepatitis B virus (HBV) infection remains a major health problem despite an extensive vaccination program worldwide. Globally, 260 million people are chronically infected with HBV and 890,000 are dying yearly from complications due to the advancement of HBV infection [1,2]. HBV may play a role in the pathogenesis of chronic liver disease, cirrhosis, and hepatocellular carcinoma (HCC). About one third of the global population is infected with HBV at some point in their lives, and the self-limiting acute hepatitis B infection ends once the virus is cleared [3,4]. When HBV is acquired in adulthood, about 5% persists and progresses into chronic hepatitis B infection (CHB) while in 95% and 20–30% cases of chronic liver disease, cirrhosis, and hepatocellular carcinomas (HCC), it is acquired in infancy and childhood, respectively [5,6]. It is one of the major blood-borne viruses, mostly transmitted through blood and body fluids, and its incubation period is around 2–5 weeks [7]. It infects and replicates in hepatocytes (liver cells), which results in severe diseases, such as cirrhosis and hepatocellular carcinoma [8]. The majority of the hepatitis B endemicity in low–medium resource countries (LMIC) is caused by the transmission of the disease from mother to child [9]. Other additional factors for HBV transmission are intravenous (IV) drug abuse, occupational exposure to infected blood products, having multiple sexual partners, and lack of awareness [10,11].

HBV is a partially double-stranded hepatotropic enveloped DNA virus and belongs to the family Hepadnaviridae [12]. It is the etiological factor for both acute and chronic hepatitis B infection in humans. Even though the occurrence of HBV infections is decreasing due to vaccination and use of antiviral therapy (by reducing the viral load of chronically infected patients), around still 3.5% of the global population is chronically infected with HBV [3]. In 2016, the WHO Global Hepatitis Health Sector Strategy articulated the elimination of viral hepatitis by 2030 [13]. As per this strategy, the target of the elimination of viral hepatitis can be achieved by increasing diagnosis for timely testing, care, and treatment [14,15]. In this article, we discuss the current challenges in diagnosing HBV and recent updates in the field.

## 2. HBV Genome

In 1965, Blumberg et al. discovered an antigen known as an “Australian antigen” in an Australian aborigine, which was later termed as the hepatitis B surface antigen [16]. The HBV particles were first visualized under an electron microscope in 1970 by Dane and colleagues. In a serum of HBV-infected patients, three types of HBV particles were observed, out of which two spherical structures had 42 and 22 nm diameter and the third filament-like structure had 22 nm diameter with variable lengths [17,18]. The larger spherical structure of 42 nm diameter, termed as the Dane particle, is the infectious virion. It consists of a lipid membrane with three viral surface antigens (HBs) which surround a nucleocapsid composed of hepatitis B core protein (HBc), viral polymerase (Pol), and viral genome DNA. The 3.2 kb genome of HBV is comprised of circular partially double-stranded DNA. This relaxed-circular DNA (rcDNA) consists of a closed (−) strand and an open (+) strand. The HBV genome encodes 4 overlapping open reading frames (ORFs), namely C, P, S, and X. The functional proteins of HBV are encoded by these overlapping ORFs as follows:ORF C: 22-kDa precore protein (p22cr), HBc and HBeAg;ORF P: viral polymerase (Pol);ORF S: viral surface antigen, L(Large)-HBs, M(Middle)-HBs, and S(small)-HBs;ORF X: HBV X protein (HBx).

Hepatitis B virus e antigen (HBeAg) is translated from an HBc reading frame and is an indicator of cccDNA replication. The rcDNA of HBV, on entering the cells, is converted into covalently closed circular DNA (cccDNA), which generates viral RNAs of different lengths due to the initiation of transcription from different promoters. The lengths of these RNAs are around 3.5 Kb, 2.4 Kb, 2.1 Kb, and 0.7 Kb. In 1983, Rall et al. showed that transcription from the HBV genome is facilitated by RNA polymerase II of an infected host and controlled by 4 different promoters and 2 enhancers (Enhancer I and Enhancer II), for preS1, preS2, core, and X sequences [19].

## 3. Genotypes of HBV and Their Clinical Importance

The HBV genotypes have been classified based on 8% or more divergence in the HBV genome sequences. To date, around 10 genotypes (A to J) along with some subtypes have been identified and their prevalence varies in different geographical regions. The circulation of particular HBV genotypes in a particular population may have some epidemiological importance, can reveal the country of origin, and can help in tracking the pattern of transmission. The HBV genotypes of immigrants are mostly traced to their native countries [20].

Globally, four HBV genotypes (A, B, C, and D) are frequently found, but the presence of the genotype B and C is common in Eastern and Western Asia, genotypes A and D in North America, Africa, and Europe, and genotype E in West Africa [21,22]. The genotype of HBV may be linked with disease progression, outcome in the chronicity of the infection, and response to the therapy. The patients infected with different HBV genotypes may show distinctive disease progression, distinctive outcomes, and antiviral therapy response [23,24,25]; however, the approved HBV vaccines are effective against all genotypes of HBV [21,25].

## 4. The Natural History of HBV Infection

The pathogenicity of HBV infection is governed by the immune response of the host interaction, virus replication, evolution, and environmental factors. For chronic HBV infection, the age at which a person acquires the infection plays an important role. The risk of progression of acute to chronic HBV infection is around 95% during the perinatal period, 20–30% in 1–5-year-old children, and less than 5% in adults [6,26]. Chronic HBV infection can be categorized into four phases (Table 1).

### 4.1. HBeAg-Positive Chronic Infection or Immune-Tolerant Phase

This is a high-replicative, low-inflammatory phase, characterized by high viral loads. Typically, there are high levels of HBV DNA (generally > 10^7^ IU/mL), positivity for HBeAg and HBsAg, but normal alanine aminotransferase (ALT) levels and normal liver histology [6,27].

### 4.2. HBeAg-Positive Chronic Hepatitis or Immune-Active Phase

This phase occurs as a result of the host immune response against HBV, resulting in liver cell injury. This manifests as elevated ALT levels together with the sign of moderate to severe liver injury. HBV replication is reduced and HBV DNA, HBeAg, and HBsAg levels decline. This phase ends with the reduction of HBV DNA and HBeAg seroconversion to anti HBe positivity [3].

### 4.3. HBeAg-Negative Chronic Infection or Immune Control Phase

This phase is characterized by HBeAg seroconversion to antibody to HBeAg (anti-HBe), low to undetectable HBV DNA, and normal ALT level. However, about 10–30% of HBeAg seroconversion patients still have elevated levels of ALT and HBV DNA; consequently, they are classified as HBeAg-negative CHB patients [6] and mostly have a mutation in the core promoter or pre-core region (antibody HBe, only core).

### 4.4. HBeAg-Negative Chronic Hepatitis or HBeAg-Negative Immune Reactivation Phase

Recurrence of replication of HBV DNA is seen in around 10–20% of inactive carriers after years of quiescence. Most of them have a mutation in the core promoter or pre-core region and show necroinflammation and fibrosis in liver histology [6].

### 4.5. Acute HBV Infection

Acute HBV infection is a condition where an increased level of HBsAg and alanine aminotransferase (ALT) is observed due to HBV infection and clears in less than 6 months. The elevated level of HBsAg and ALT were eliminated within 6 months. Acute HBV infections are mainly asymptomatic and only 30% of infected persons showed clinical signs of jaundice and hepatitis [28]. Acute HBV infection has an “eclipse” period of around 8 days, in which no evident phase of infection is observed [29]. Acute HBV infections are self-limiting but can persist as a residual infection and can be active in immune-compromised individuals. Initially, it was assumed that the DNA of HBV is eradicated but sensitive PCR assays may be able to detect DNA traces from serum and liver [30]. HBV infections that cannot clear within 6 months can lead to the chronic HBV infections. Chronic HBV infection is characterized by a high levels of HBsAg and ALT due to weak cytotoxic T-cell response.

### 4.6. Occult HBV Infection (OBI)

OBI is defined as presence of replication competent HBV DNA (i.e., episomal covalently closed circular DNA) in the liver and/or HBV DNA in the blood of persons having undetectable HBsAg by presently available tests [31,32].

In OBI, there is suppression of viral replication activity and protein expression, due to host’s immunologic and epigenetic mechanisms, resulting in absence of HBsAg. OBI can be categorized as:Sero-positive-OBI: positive for anti-HBc and/or anti-HBs antibodies;Sero-negative-OBI (1–20% of all OBIs): Negative anti-HBc and/or anti-HBs antibodies.

Missing out on HBsAg positivity due to the inadequate sensitivity of assay or inability to detect HBV S variants may lead to false negative HBsAg and misdiagnosis/false positive OBI. Though HBsAg is negative, episomal cccDNA in OBI cases is fully replication competent (unlike integrated HBV DNA) and can lead to a reactivation of infection in patient or transfusion-transmitted hepatitis B virus infection in a recipient of blood from an OBI positive donor. Diagnosis of OBI depends on detection of HBV DNA in the liver (gold standard) or blood (more commonly used), with absence of HBsAg. Presence of anti HBcAb is often used as a surrogate marker. Detection of HBV DNA in blood is challenging as it is present in low concentrations and may only be intermittently detected. Hence, the use of sensitive methods for detection of low levels of HBV DNA is recommended. The presence of anti HBc with absence of HBs Ag and presence of anti HBs is also seen in recovered patients, however, HBV DNA is negative in such patients [29,31,32]. A resolved HBV infection is defined as a positive HBc antibody without detectable serum HBV DNA or negative HBsAg.

## 5. Diagnosis of HBV

WHO’s plan to eliminate viral hepatitis by 2030 can be accomplished by increasing diagnosis, care, and treatment [13]. The elimination of hepatitis refers to the reduction of 90% incidence and 65% of deaths from the 2015 baseline [14,15,33]. Screening and diagnosis should be made available to the people unacquainted with their status of HBV infection or who have not so far entered into care and treatment [34,35]. These unaware infected people recurrently carry on a spread of the virus [36].

The diagnosis and follow-up of chronic infection rely on laboratory viral biomarkers. There are two key categories of HBV biomarker assays: one is serology, a term encompassing the detection and quantification of viral-specific antibodies and/or antigens, and the second is nucleic acid testing (NAT) for the detection and quantification of the HBV genome and its RNA transcripts [28].

Serology tests that identify or measure HB surface antigen (HBsAg) serum levels, HB surface antibodies (anti-HBs), and HB core antibodies (anti-HBcs) are used to detect patients who have been exposed to HBV, whereas NAT tests provide information on the level of virus replication, the manifestation of particular variants, and occurrence of virus reservoirs. Tests are being advanced to measure levels of intrahepatic HBV replication. These biomarkers are to be used to identify patients with HBV infection, follow disease progression, and determine response to therapy and efficacy of new agents in clinical trials.

HBV can be diagnosed by various HBV markers such as hepatitis B surface antigens (HBsAgs), hepatitis B surface antibodies (anti-HBs), hepatitis B e-antigen (HBeAgs), hepatitis B e-antibodies (anti-HBes), and hepatitis core antigens (anti-HBcs) (Table 2)

### 5.1. Routinely Used HBV Markers

Hepatitis B surface antigen (HBsAg): the key serological marker for acute and chronic hepatitis B infection. HBsAg is used as a marker to establish the prevalence of chronic HBV infection in epidemiological studies [37]. Usually, detection of the HBsAg in repeat testing (after 6 months) is used as marker of chronic HBV infection, whereas the absence of HBsAg in serum indicates recovery from acute HBV infection [38].

It is encoded by ORF S and synthesized as a small protein (SHBs), a medium protein (MHBs), and a large protein (LHBs) viral surface antigen [39]. Viral transmission occurs when SHBs and/or LHBs which contain the pre-S1 region binds to the sodium taurocholate co-transporting polypeptide and heparan sulphate proteoglycan (Urban S, et al., 2014). HBsAg is part of the viral envelope and also occurs as a non-infectious subviral particle (which may inhibit the host immune system) and its measurement in body fluid has been used as an important diagnosis parameter in clinical practice and trials [40,41].

HBsAg is the key marker of chronic HBV infection when persistence is more than 6 months, whereas after recovery from acute HBV infections, levels of HBsAg become undetectable.

The levels of HBsAg in the body fluid are important to reveal the infection stage because it reflects the transcriptional activity of cccDNA, which is higher in HBeAg-positive infection than in HBeAg-negative patients [42,43].

HBsAg quantification has prognostic significance and has been incorporated into risk scores to predict the risk of HCC and possibly specify rebound viral risk after stopping NUCs [40]. Currently, commercially available standardized assays for the quantification of HBsAg are the Architect HBsAg assay developed by Abbott Diagnostics, USA, the ElecsysHBsAg II quant assay developed by Roche Diagnostics, USA, and the DiaSorin Liaison XL, developed by DiaSorin, Italy. These assays can detect and quantify the HBsAg but cannot differentiate three HBs proteins. These three HBs proteins can be detected and distinguished by in-house ELISA or Western blot analysis [44].

Anti-HBs (Antibody to HBsAg): the main marker to show the presence of antibodies that neutralize HBV and generally symbolize recovery from an acute infection. After recovery, anti-HBs and anti-HBc may be detectable. A positive anti-HBs with negative HBsAg can be seen in response to HBV vaccination, recovery after acute hepatitis, or HBsAg seroconversion in chronic HBV infection. The anti-HBs titer of less than 10 IU/l is considered negative, whereas 10–100 IU/l is considered moderate. The titer higher than 100 IU/l is considered protective even if the person is exposed to a high HBV viral load [28,29].

Hepatitis B core antigen (HBcAg) and anti-HBc: the core protein is covered with HBsAg and thus not freely detected in serum. Few assays are available for anti-HBc but not for detection of HBc antigen [29]. The detection of Anti-HBc is considered to be suggestive of HBV contact and predictive of three outcomes:Recovery indicated by the development of anti-HBs;Chronicity indicated by long-term HBsAg;Occult HBV infection (OBI) is indicated by low levels of HBV DNA and absence of HBsAg.

In acute HBV infection, the first detectable antibody is IgM anti-HBc, which can be detected within 1 month after HBsAg appearance. The recent infection shows elevated levels of IgM anti-HBc and it lasts for around 4–6 months. During recovery from the acute infection, the levels of IgG anti-HBc increase whereas IgM anti-HBc levels decrease. Low levels of the IgM anti-HBc can persevere in HBV chronic infection and can increase with the severity of chronic hepatitis B. Detection of anti-HBc in the blood may be used as a surrogate marker for identifying OBI in blood or organ donors, in persons who are about to receive immunosuppressive therapy, and for epidemiological studies [29].

Hepatitis B e antigen (HBeAg) and anti-HBe: HBeAg is coded by the precore (Pre-C) region of the core gene (C). HBeAg helps in distinguishing HBeAg-positive and HBeAg-negative CHB infection. HBeAg seroconversion marks the transition from the immune clearance phase to the immune control phase of CBH. HBeAg has been used as a marker of chronic active HBV infection in its immune tolerant phase and reactivation of low-replicative chronic infection [45,46]. The existence of HBeAg in the serum of an HBsAg-positive carrier indicates frequent viral replication and greater infectivity. Assays for detecting HBeAg and anti-HBe are generally combined in the same enzyme immunoassay kits as these markers are essentially mutually exclusive. HBeAg can be used as a cost-effective substitute for HBV DNA, which help in therapeutic management and efficacy monitoring against HBV in low resource countries [47,48].

Hepatitis B virus DNA (HBV DNA): testing of the DNA in serum is used to assess HBV viral replication and should be performed regularly, around 6 months in chronic HBV patients. The viral load of HBV DNA provides insight for current therapy guidelines and treatment efficacy [6,45]. The viral cccDNA was not usually used to detect HBV, though used to quantify the viral load [49,50]. The occurrence of the cccDNA in liver cells is a critical cause of difficult HBV eradication as cccDNA acts as a template for the new virions replication [51]. The absence of HBsAg and cccDNA indicates the true HBV cure [26]; however, an increase in HBV DNA concentration indicates resistance to given therapy [3].

### 5.2. Emerging Marker for HBV

Newer markers are continuously being sought for better diagnosis, prognostication, and treatment management. Recent markers include HBcrAg and HBV RNA.

Hepatitis B core-related antigen (HBcrAg): chronic HBV infection cannot be eliminated due to the presence of cccDNA in the nucleus of infected liver cells. The biopsy of liver cells, required to quantitate the cccDNA, is a difficult and invasive procedure. The serological marker, HBcrAg, may be an alternative non-invasive marker for intrahepatic viral replicative activity. Some studies showed a good association between the amount of HBcrAg and cccDNA in both HBeAg negative and positive patients [52,53,54] but a feeble association with HBsAg [55]. HBcrAg comprises of three proteins HBeAg, p22cr, and HBcAg, which are coded by the precore/core region and can be used in serologic testing [55,56,57]. These three proteins share an identical 149 amino acid sequence and are detectable when HBV DNA and HBsAg are undetectable [58,59]. HBeAg, encoded by a core gene, is a circulating peptide transformed and secreted in hepatocytes. HBcAg nucleocapsid protein surrounds viral DNA whereas p22Cr exists in HBV DNA and HBcAg negative Dane-like particles [58]. HBcrAg is a prospective alternate indicator of cccDNA and may soon turn into advantageous marker for outcome and management of the HBV related infections or diseases [60].

Hepatitis B virus RNA (HBV RNA): transcripts (HBV mRNA) act as a template for synthesis of viral proteins, therefore, mRNAs and pregenomic RNAs can act as a viral replication markers which are present in the serum of infected patients [61]. Interestingly, many commercial assays are available for the detection of HBV DNA, but no commercial assay is available to detect HBV RNA. In absence of commercial assays, in-house assays may be used to detect HBV RNA that correlates with HBV DNA in untreated patients. It is important to note that detection of HBV RNA may be influenced by variables such as the genotype and presence of mutations. Similar assumptions can be made about other diagnostic markers. The knowledge of these variables may become advantageous for future studies to make HBV RNA detection assays more reliable [62,63].

The HBV biomarkers, such as the hepatitis B surface antigen (HBsAg), hepatitis core antigen (anti-HBc), hepatitis B e-antigen (HBeAg), hepatitis B surface antibody (anti-HB), and hepatitis B e antibody (anti-HBe), have been used for the detection the natural infection and infection phases of HBV. However, the levels of HBV DNA play a vital role in management of HBV infection [64].

## 6. Common Methods for Detection of Hepatitis B Virus

The detection and quantification of the hepatitis B viral markers in body fluids is mostly carried out by enzyme-linked immunosorbent assays (ELISA), radioimmunoassay (RIA), enzyme immunoassay (EIA), polymerase chain reaction (PCR), and recently developed techniques such as microparticle enzyme immunoassay (MEIA), electrochemiluminescence immunoassay (ECLIA) and chemiluminescence microparticle immunoassay (CMIA). The advanced techniques for HBV testing are often expensive and need bulk instrumentation and trained manpower [15,65,66]. The advantages and disadvantages of the detection methods of HBV are given in Table 3.

### 6.1. Point-of-Care Tests (POCT)

The fast and more accessible diagnosis of the HBV is aided by POCTs, also known as rapid diagnostic tests (RDT). These tests require 1–2 drops of sample and are easy to use, and do not require specialized training, which makes them ideal for variety of community and outreach locations. The sensitivity of these tests kits are less in comparison to other tests [65].

### 6.2. Dried Blood Spot (DBS)

A sampling method which aids for practical solution for the large population screening or testing in low-resource settings having limited accesses to the testing facility. A drop of patient blood by finger-prick is collected on a chemically treated paper card. The chemical on the paper card preserves the HBV marker during transport at ambient temperature from field to laboratory, where these samples are tested using advanced molecular or immunoassays [67,68].

## 7. Challenges in HBV Diagnosis

In low-resource settings or countries, collecting blood for HBV testing at a remote location usually requires additional logistic support for transport to the testing facility. HBV diagnosis usually relies on plasma or serum for most commonly available tests. Low-resource setting sub-centers usually lack centrifugation machines needed to separate plasma and serum from blood samples. Some sub-centers may have centrifugation machines, but it is difficult to arrange supplies such as consumables and power. Samples transported at room temperature do not usually affect serological test markers, but can degrade molecular markers. This issue can be overcome by using the DBS method of sample collection [69,70,71,72] and using point-of-care tests. These measures may help in overcoming the lack of testing and loss of follow-up in low resource settings [73]. Nucleic acid testing for screening HBV is being used extensively in the developed countries but it has limited use in low-income countries due to a higher cost and requirement of trained manpower.

## 8. Future Perspectives

The proper understanding and interpretation of diagnostic methods is necessary for successful therapy against hepatitis B, because different type of therapies target various markers of hepatitis B infection differently. Therefore, diagnosis based on different combinations of markers of hepatitis B infection needs to be carried out to monitor the effectiveness of a therapy. The seroclearance of HBsAg and appearance of anti-HBs [26,45], which further indicates protective immunity, can be diagnosed with a combination of HBs and anti-HBs markers. HBsAg serum levels and their source are different in the inactive and immune–tolerant phase and, therefore, need to be diagnosed by different strategies in these phases. In HBeAg-negative patients, less than 100 IU/mL of HBsAg may indicate gradual HBsAg clearance that can only be diagnosed together with HBeAg and HBsAg markers [74]. Both markers, along with knowledge of HBV genotypes, may also be helpful in monitoring the response of antiviral therapy. The genotypes of HBV in the infected patient is known to influence the effect of antiviral therapy [75]. Therefore, the foolproof diagnosis for HBV would be to obtain information on HBsAg, HBeAg, hepatitis B viral load, and genotypes of HBV. These four markers may help in accomplishing the WHO’s goal because these will help in determining the stage of infection and further decision-making in the type of therapy to be given.

## 9. Conclusions

In spite of the fact that there are several unresolved and controversial issues in the diagnosis of hepatitis B, there is a necessity for mass testing, especially in areas where hepatitis B is more prevalent and where resources are limited. There is, however, an urgent need for the development of sensitive, standardized, and validated test procedures for the detection of HBsAg, HBV DNA (in blood and in liver), and other viral markers. These tests should identify HBV S variants as well as HBsAg present within immune complexes with anti-HBs. A standard report on occult blood infections should be adopted, so that this type of infection can be reported in the future. As part of the WHO’s goal to eliminate viral hepatitis from the public health agenda by 2030, the improvement of awareness of the disease, case identification, surveillance strategies, and treatment optimization are crucial steps towards achieving this goal.

## Figures and Tables

**Table 1 diagnostics-13-00368-t001:** Phases of Chronic HBV Infections.

Phase Characteristics	HBeAg-Positive Chronic Infection	HBeAg-Positive Chronic Hepatitis	HBeAg-Negative Chronic Infection	HBeAg-Negative Chronic Hepatitis
HBV DNA	Very high	High	Low or undetectable	High
ALT	Normal	Elevated	Normal	Elevated
HBsAg	Detected	Detected	Undetected	Intermediate
Liver Histology	Normal	Moderate	Minimal	Moderate to severe
HBe Ag	Positive	Positive	Negative	Negative

**Table 2 diagnostics-13-00368-t002:** HBV markers and their significance in diagnosis.

Marker	Definition	Significance in Diagnosis
HBV DNA	Indicator of active HBV infection, assess viral replication	Can be detected in window period Identifies occult HBV infection (OBI) in patients with liver disease or co-infection Conc. of HBV provides an indication for therapy and monitoring treatment efficacy
HBsAg	Hallmark of infection	First serologic marker to appear Persists for more than 6 months in chronic HBV infection Important for the knowing the infection stage and reflect transcriptional activity
Anti-HBs	Neutralizing antibody	Recovery and/or immunity to HBV The only marker detectable after immunity conferred by HBV immunization
HBeAg	Indicator of active replication of HBV	Sero-status reflection of natural history of disease Markers to distinguished HBeAg positive chronic infection and hepatitis from HBeAg negative chronic infection Marker of reactivation of low-replicative chronic infection
Anti-HBc	Indicates exposure to HBV	Indicates the contact with HBV HBV reactivation after potent immune-suppression in HBsAg-negative phase of disease
Anti-HBe	Found early in the course of acute hepatitis B and disappears soon after ALT peaks	Less active HBV replication Decrease of HBV infectivity and remission of disease Precore/core promoter mutations
IgM anti-HBc	Recent infection with hepatitis B virus	Presence with high index value during acute HBV infection and usually disappears within 6 months 10–20% of chronic hepatitis B patients with hepatitis flares also positive for IgM anti-HBc with low index value
IgG anti-HBc	Indicates a chronic infection	Not a neutralizing antibody and presence indicates an exposure to HBV Isolated IgG anti-HBc may indicate OBI

**Table 3 diagnostics-13-00368-t003:** Advantages and disadvantages of the HBV diagnostic methods.

Methods	Advantages	Disadvantages
EIA	This is an automated process that generates high reproducible results at a low cost	This is a time-consuming process because it requires sophisticated equipment and trained technicians. It also requires ongoing supply of electricity, which is not suitable for use in the field
RIA	Sensitivity is very high	There is a high cost associated with this method There is a risk for the operator
MEIA	An immunological method with high sensitivity and a faster turnaround time than any other methods	A high level of sophistication of equipment, trained technicians, and constant electricity supply
ECLIA/ CMIA	An automated system with a high degree of specificity and sensitivity	high cost, sophisticated equipment, experienced technicians, and continuous power supply
PCR	Capacity of detecting low viral loads; Broad dynamic range; Less chances of carry over contamination; Can be fully automated	A high level of sophistication of equipment, trained technicians, and constant electricity supply

## Data Availability

Not applicable.

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
