# Peer review of "Current Challenges and Future Perspectives of Diagnosis of Hepatitis B Virus"

_diagnostics, 2023, doi:10.3390/diagnostics13030368_

Round 1
Reviewer 1 Report
Authors summarized the current available techniques for diagnosis of Hepatitis B virus. Overall, the context is important for chasing the goal of eliminating this virus. Some revisions are required to improve the manuscript.
The pros and cons of each diagnostic method should be discussed in a table.
Future perspectives are not really discussed in the manuscript.
Abbreviations, such as WHO, HBeAg, should be defined as the first time shown in the manuscript.
The space and punctuations should be checked across the manuscript, such as before or after references (studies. [37].) and Marker ofReactivation.
Author Response
Reviewer 1:
- The pros and cons of each diagnostic method should be discussed in a table.
Response: As suggested we have incorporated in Table 3
- Future perspectives are not really discussed in the manuscript.
Response: As suggested by the reviewer “Future perspectives” has been added above the conclusion section in the manuscript
Future perspectives
Diagnostic methods must be understood and interpreted correctly to ensure successful treatment of hepatitis B because different therapeutic approaches target different markers of infection differently. In order to monitor the effectiveness of hepatitis B therapy, different combinations of markers of infection need to be assessed. In addition to HBsAg seroclearance and anti-HBs appearance [26, 45], combination of anti-HBs and HBs markers can be used to diagnose further acquisition of protective immunity. Inactive and immune-tolerant phases of HBsAg have different serum levels and sources, which makes them susceptible to different diagnostic strategies. The presence of HBsAg less than 100 IU/ml in patients without HBeAg may indicate gradual clearance of the antigen [74] and such a process can only be detected with both HBeAg and HBsAg markers. It is also possible to monitor antiviral treatment response using both markers and genotype information. Antiviral treatment is known to be affected by the genotypes of HBV in the infected patient [75]. Therefore, HBsAg, HBeAg, viral load, and genotype of HBV are all necessary for a foolproof diagnosis of HBV. By identifying these four markers, WHO will be able to determine the stage of infection and more effectively determine the type of therapy to be given thereby accomplishing the goal.
- Abbreviations, such as WHO, HBeAg, should be defined as the first time shown in the manuscript.
Response: The authors are thankful to the reviewer for highlighting this and corrected as per suggestion
- The space and punctuations should be checked across the manuscript, such as before or after references (studies. [37].) and Marker of Reactivation.
Response: The authors are thankful to the reviewer for highlighting this and corrected as per suggestion

Reviewer 2 Report
It is an important study in the field of virology and hepatology. The section on recent diagnoses, including pathological methods, needs to be discussed in detail. Previous studies discussed this issue in detail; what is new with your research?
Author Response
- It is an important study in the field of virology and hepatology. The section on recent diagnoses, including pathological methods, needs to be discussed in detail. Previous studies discussed this issue in detail; what is new with your research?
Response: We thank for reviewer concern and suggestion. We have not discussed these things in detail because our study is mainly focused on diagnostic markers (serology part mainly). Describing other things will lead to deviate the main theme of the manuscript and also it may become bulky with less interest of audience.

Round 2
Reviewer 1 Report
Great job. No additional comments.
Author Response
thanks